# The Role of the NLRP3 Inflammasome and Programmed Cell Death in Acute Liver Injury

**DOI:** 10.3390/ijms24043067

**Published:** 2023-02-04

**Authors:** Chaoqun Yu, Peng Chen, Longyu Miao, Guohu Di

**Affiliations:** 1School of Basic Medicine, College of Medicine, Qingdao University, Qingdao 266071, China; 2Institute of Stem Cell and Regenerative Medicine, School of Basic Medicine, Qingdao University, Qingdao 266071, China

**Keywords:** acute liver injury, NLRP3 inflammasome, programmed cell death

## Abstract

Acute liver injury (ALI) is a globally important public health issue that, when severe, rapidly progresses to acute liver failure, seriously compromising the life safety of patients. The pathogenesis of ALI is defined by massive cell death in the liver, which triggers a cascade of immune responses. Studies have shown that the aberrant activation of the nod-like receptor protein 3 (NLRP3) inflammasome plays an important role in various types of ALI and that the activation of the NLRP3 inflammasome causes various types of programmed cell death (PCD), and these cell death effectors can in turn regulate NLRP3 inflammasome activation. This indicates that NLRP3 inflammasome activation is inextricably linked to PCD. In this review, we summarize the role of NLRP3 inflammasome activation and PCD in various types of ALI (APAP, liver ischemia reperfusion, CCl_4_, alcohol, Con A, and LPS/D-GalN induced ALI) and analyze the underlying mechanisms to provide references for future relevant studies.

## 1. Introduction

The liver is the largest parenchymal organ in the human body and is the key hub of many physiological processes, including nutrient metabolism, blood volume regulation, immune regulation, and detoxification [1,2]. As such, the normal physiological structure and function of the liver are essential for maintaining human health [3]. Liver failure, due to various factors, is a global health problem. Acute liver injury (ALI) is a common disease that seriously threatens the health of patients, and the massive death of hepatocytes in patients within a short period of time leads to a dramatic decline in their liver function [4]. The etiology of ALI can generally be defined as either drug-induced liver injury, ischemia-reperfusion liver injury, autoimmune liver injury, or viral liver injury [5,6]; viral causes predominate in developing countries while drug causes predominate in developed countries such as the United States and Western Europe [7]. ALI progresses rapidly, and once acute liver failure has occurred, liver transplantation is the only effective treatment [8]; however, the scarcity of liver sources, the cost, and high mortality rates after urgent transplantation limit its application [9]. Various causes of innate immunity drive a local sterile inflammatory response and are the most direct cause of liver injury and failure [10], so it is particularly important to better understand the immune response in ALI and inhibit its progression.

The nod-like receptor protein 3 (NLRP3) inflammasome is the most widely studied of the inflammasomes. It is composed of a sensor (NLRP3), an adapter (apoptosis-associated spotted protein, ASC), and an effector (cysteine aspartate protease 1, caspase-1) [11,12]. The level of NLRP3 expression under steady-state conditions is insufficient for the activation of the NLRP3 inflammasome [13]. Thus, its activation requires two steps [14,15,16]. Signal 1 (pathogen-associated molecular patterns (PAMPs) and cell factors) stimulates toll-like receptor 4 (TLR4), activating nuclear factor kappa-B (NF-ĸB) and promoting its translocation to the nucleus [17,18]; this then upregulates the expression of NLRP3, pro-interleukin-1β (IL-1β), and pro-interleukin-18 (IL-18) [19]. Signal 2 (damage-associated molecular patterns (DAMPs) and PAMPs) then induces NLRP3 protein oligomerization, recruits the ASC and pro-caspase-1 molecules [20], and then interacts with them to assemble the NLRP3 inflammasome, activating caspase-1 [15,21]. On the one hand, activated caspase-1 promotes the maturation of proinflammatory cytokines, such as IL-1β and IL-18 [22]. On the other hand, activated caspase-1 also cleaves gasdermin D (GSDMD) and releases its N-terminal domain, which translocates to the cell membrane and forms pores, leading to the release of cellular contents such as IL-1β and IL-18, inducing pyroptosis (Figure 1) [13,15,23,24]. Studies have shown that the NLRP3 inflammasome plays an important role in various diseases throughout the body, including chronic liver injury (viral hepatitis, non-alcoholic fatty liver disease, etc.) and ALI [25,26,27,28,29,30,31,32].

Biologists have long regarded cell death as an inevitable, passive outcome of life. However, as research has developed over the past few decades, studies have shown that organisms can actively clear damaged, harmful, and superfluous cells through certain specific cell death programs; this process is essential for maintaining the homeostasis of the human body environment [33,34]. Depending on whether the cell death process can be regulated, it can be classified as either accidental cell death (ACD) or regulated cell death (RCD) [35]. ACD is uncontrollable, mainly because the cells are subjected to extreme physical, chemical, or mechanical stimuli (e.g., high temperature, high pressure, pH changes, and shear forces). However, RCD is induced by precise molecular mechanisms that intervene in regulated cell death at the biological level [36]. RCD that occurs under physiological conditions that lack any exogenous environmental disturbance is also known as programmed cell death (PCD); this category includes apoptosis, necroptosis, pyroptosis, ferroptosis, and autophagy, among other forms [37,38,39,40]. In recent years, intensive research into the molecular mechanisms at play during cell death has shown that multiple effectors of PCD can regulate the activation of the NLRP3 inflammasome and that the activation of the NLRP3 inflammasome can also lead to different forms of PCD, indicating that PCD is inseparable from the activation of NLRP3 inflammasome [41,42,43]. In addition, with the rapid development of single-cell sequencing technology, several results of liver transcriptomic analysis have further demonstrated that NLRP3 inflammasome is closely related to various regulatory and effector factors of PCD [44,45,46,47]. In this review, we focused on current studies regarding the NLRP3 inflammasome and PCD in ALI and analyzed the underlying mechanisms to provide references for future research and treatment of ALI.

## 2. NLRP3 Inflammasome and Programmed Cell Death

PCD is an active, orderly mode of cell death that is encoded in specific genes. Different types of PCD differ in their morphological features, signaling both the pathways and their resulting outcomes; here, we will mainly discuss the characteristics of the following types of PCD and their mutual regulation with the NLRP3 inflammasome: apoptosis, necroptosis, pyroptosis, and autophagy.

### 2.1. Apoptosis

As the first to be defined in the literature, apoptosis is the most classic form of PCD [48,49,50]. Apoptosis was first observed in 1842 by Karl Vogt during cell growth and development in toads and was then formally described by Kerr in 1972 [51,52]. The main morphological characteristics of apoptosis are cell deformation, loss of cell membrane integrity, DNA degradation, and the formation of apoptotic bodies (Figure 2) [53]. According to their inducing factors, cases of apoptosis can be divided into cell mitochondria/caspase-9 mediated endogenous apoptosis and death receptor/caspase-8 mediated exogenous apoptosis [54,55,56]. Although the initiation signaling pathways vary, all cases share common downstream pathways that culminate in the execution of apoptosis through a family of caspases (Caspase-3, -6, -7) [41,57,58]. Studies have shown that the family of caspases, which are initiators and executors of apoptosis, can promote the activation of the NLRP3 inflammasome by inducing potassium ion efflux [59,60]. In addition, several different effectors of apoptosis, such as the Bcl-2-associated X protein and Bcl-2 homologous antagonist killer, can regulate mitochondrial homeostasis and thereby induce NLRP3 inflammasome activation [61]. However, there are also some effectors of apoptosis that have diametrically opposite effects; for example, cytochrome c released from the inner mitochondrial membrane upon apoptosis can competitively bind to the LRR domain of NLRP3, thereby inhibiting its activation [62,63]. The intrinsic link between apoptosis and NLRP3 inflammasome activation is complex, and so a careful distinction between the two is required when judging the relationship between apoptosis and NLRP3 inflammasome in disease.

### 2.2. Necroptosis

Necroptosis is another type of PCD that occurs when the normal apoptotic pathway of cells is inhibited [64,65]. In 1998, Vercammen et al. found that fibrosarcoma cells induced a controlled form of necrosis in the absence of caspase [66]. In contrast to apoptosis, necroptosis does not affect karyopyknosis, nor produce nucleosome DNA fragments, which are mainly manifested through mitochondrial dysfunction, organelle disruption, and leakage of cell contents [67,68]. According to various inducing factors, necroptosis is divided into tumor necrosis factor-α(TNF-α)-induced external necroptosis, intrinsic necroptosis induced by reactive oxygen species (ROS), and endogenous necroptosis induced by ischemia [69]. Of these, TNF-α-mediated necroptosis is the most prevalent form in the literature and is widely studied [64,70]. The activation of receptor-interacting serine-threonine kinase 3 (RIPK3) and mixed lineage kinase domain-like (MLKL) is key in the necroptosis pathway, and studies have shown that they can promote NLRP3 inflammasome activation through multiple pathways [71,72,73,74]. In addition, Z-DNA binding protein 1 (ZBP1), a recently discovered effector of necroptosis, has also been shown to play an important role in NLRP3 inflammasome activation [75,76].

### 2.3. Pyroptosis

Pyroptosis is a type of programmed necrosis that is dependent on the gasdermin family [37]. In 1992, Zychlinsky et al. discovered pyroptosis in *Shigella flexneri*-induced macrophages, but incorrectly classified it as a specific category of apoptosis [77]. It was not until 2001 that Cookson formally proposed the concept of pyroptosis, defined as caspase-1-dependent inflammatory programmed cell death [78]. It is mainly characterized by cell swelling, pyknosis of the nucleus and DNA fragmentation, and the rupture of the cell membrane to form pores that then release a large number of inflammatory factors [79,80]. The occurrence of pyroptosis is mainly divided into the caspase-1-dependent canonical pathway and the caspase-4, -5, -11-dependent non-canonical pathway [79,81,82,83]. Pyroptosis is the most closely related form of PCD to the NLRP3 inflammasome, as the canonical pyroptosis pathway can be initiated by caspase-1 and the inflammasome initiates caspase-1 activation, of which the NLRP3 inflammasome is currently the most extensively studied [84].

### 2.4. Autophagy

Autophagy is a lysosome-dependent phagocytic phenomenon that is genetically regulated and is also considered a type II PCD [85]. Autophagy plays a role in the energy cycle and in maintaining cellular homeostasis by degrading proteins and damaged organelles inside the cell [86,87,88,89,90]. The primary morphological feature of autophagy is the formation and accumulation of either crescent-like or cup-shaped phagocytic vacuoles [91,92]. Autophagy can be divided into non-selective autophagy, such as macroautophagy and microautophagy, and selective autophagy, which includes mitophagy, peroxisomal autophagy, endoplasmic reticulum autophagy, as well as ribosomal autophagy, among others [87,93]. Macroautophagy is currently the most widely studied form of autophagy; in general, the word autophagy is used to refer to macroautophagy. Autophagy is reciprocally regulated with the NLRP3 inflammasome and can either reduce DAMPs release by removing damaged organelles [94], thereby inhibiting NLRP3 inflammasome activation [95,96,97], or inhibit its activation by engulfing and degrading NLRP3 inflammasome components [98,99,100,101]. The NLRP3 inflammasome can also inhibit autophagy activation through autophagy protein interactions [102,103,104].

## 3. NLRP3 Inflammasome and Programmed Cell Death in ALI

The NLRP3 inflammasome, as well as various forms of PCD, have been shown to play important roles in the pathogenesis of ALI, and their in-depth study has contributed to the development of new therapeutic strategies for ALI. Here, we summarize both the pathogenesis of the NLRP3 inflammasome and PCD in various types of ALI and the latest therapeutic strategies targeting them.

### 3.1. NLRP3 Inflammasome and Programmed Cell Death in Acetaminophen (APAP)-Induced Acute Liver Injury

Drug-induced liver injury (DILI) is the main form of acute liver injury in developed countries [105,106]. APAP is a widely used antipyretic and analgesic drug that is safe to use at therapeutic doses [107,108]. However, when it is over-used it can produce dose-dependent hepatotoxicity and even liver failure [109]. APAP-induced acute liver injury is the leading cause of drug-induced liver injury in western countries; in both the United States and the United Kingdom, the greatest predisposition for cases of acute liver failure is the excessive use of APAP [110,111,112]. N-acetylcysteine (NAC) is the only clinically approved drug for the treatment of APAP-induced liver injury. However, the NAC therapeutic window is quite narrow and can have adverse effects [113,114,115]. At a normal dose, APAP is converted in the liver, primarily by UDP-glucuronosyltransferase (UGT) and sulfotransferase (SULT), into a nontoxic compound and excreted with urine; only a small amount is metabolized to n-acetyl-p-benzoquinoneimine (NAPQI) by cytochrome P450 enzymes (CYP) [116]. NAPQI is a highly reactive intermediate compound that is cytotoxic and can be rapidly detoxified in combination with glutathione (GSH) in the liver [117,118]. However, once GSH is depleted, NAPQI covalently binds to protein sulfhydryl groups, especially in the mitochondria, leading to mitochondrial oxidative stress, and thereby inducing hepatocytes to undergo both apoptosis and necrosis [119,120,121]. Mice are preferred for studying APAP overdose because the doses of APAP that cause toxicity are similar in mice and humans, and the mechanisms of toxicity are similar in mice and humans. However, the time course of liver injury differs slightly between humans and mice, with hepatotoxicity developing slightly faster in mice, peaking at around 12–24 h compared with 24–72 h in humans [122].

In recent years, increasing evidence has demonstrated that apoptosis plays an important role in APAP-induced liver injury; tune staining has also demonstrated that hepatocytes underwent massive apoptosis in an APAP-induced acute liver injury model in mice [123,124]. Kaempferol (KA), a flavonoid compound extracted from Penthorum chinense, has powerful anti-inflammatory and antioxidant activities. Du et al. found that KA inhibited the high mobility group protein B1 (HMGB1)/TLR4/NF-κB signaling pathway and the activation of the NLRP3 inflammasome, thereby protecting the liver from APAP-induced inflammatory responses and apoptosis [125]. Elshal et al. found that diacerein reduced NLRP3/caspase-1/IL-1β by downregulating the IL-4/Monocyte chemoattractant protein-1 (MCP-1) and TNF-α/ NF-κB inflammatory signaling pathways and mediating oxidative stress, mitochondrial dysfunction, necrosis, sterile inflammation, and apoptosis, thereby preventing and reversing APAP hepatotoxicity in mice [126]. In addition, recent studies have shown that pyroptosis and necroptosis also play a role in APAP-induced acute liver injury. Wang et al. found that targeting mitochondrial ROS (via peroxidase 3) inhibited NLRP3 inflammasome activation and prevented APAP-induced pyroptosis, thereby exerting hepatoprotective effects [127]. Receptor-interacting serine-threonine kinase (RIPK1) and RIPK3 play important roles in necroptosis, and Deutsch et al. found that in APAP mediated ALI, RIPK3 deletion or specifically blocking RIPK1 using Necrostatin-1 (Nec-1), an inhibitor of RIPK1, was protective against liver injury and associated with inhibition of NLRP3 inflammasome activation [128]. Another study suggested that Nec-1 attenuates APAP-induced liver injury by inhibiting the interaction between necroptosis and the NLRP3 inflammasome [129].

### 3.2. NLRP3 Inflammasome and Programmed Cell Death in Liver Ischemia-Reperfusion Injury

Liver ischemia-reperfusion injury (LIRI), an inevitable pathophysiological process that occurs during several clinical procedures, such as liver transplantation and liver resection, is a common type of acute liver injury [130]. Severe LIRI causes liver dysfunction and failure, which leads to surgical failure [131,132]. To date, no effective method for the prevention and treatment of LIRI has been found. Therefore, in order to provide a clinical solution to this problem, it is important to deeply investigate the pathogenesis of LIRI. LIRI can be divided into two phases, ischemia and reperfusion, which are characterized by hypoxia-induced cell injury in the ischemic phase and immunoinflammation after the restoration of blood flow [133]. During ischemia, ROS production as well as organelle damage, are induced by glycogen depletion, inadequate oxygen supply, and ATP depletion, which can lead to hepatocyte injury and death [134,135]. Subsequent reperfusion not only causes cellular metabolic disorders but also triggers a series of inflammatory cascades, exacerbating hepatocyte injury [10]. LIRI can be divided into warm ischemia and cold ischemia. Except for liver transplantation, partial hepatectomy, trauma as well as hemorrhagic shock were warm ischemia. The warm rodent model (orthotopic) LIRI model is the most widely used because of its high feasibility, few ethical concerns, and low expense [136].

There are many different pathways of PCD, all of which can initiate inflammatory immune cascades during LIRI [10]. The NLRP3 inflammasome has been shown to play an important role in LIRI pathogenesis, regulating hepatocyte injury, immune cell activation, and hepatic inflammatory responses [137,138]. Inoue et al. found that NLRP3 expression was elevated in a murine partial (70%) hepatic warm ischemia-reperfusion model, but that NLRP3^-/-^ mice showed reduced levels of inflammation and apoptosis; the authors suggested that NLRP3 regulated neutrophil function by influencing chemokine-mediated signaling [139]. γ- Oryzanol (ORY), an important bioactive ingredient isolated from rice bran oil, has anti-inflammation and anti-oxidation properties and can exert protective effects in a variety of liver disease models [140,141]. Du et al. found that ORY protected the liver from I/R-induced inflammasome activation and apoptosis by inhibiting the HMGB1/NLRP3/IL-1β signaling pathway by establishing a hepatic 70% warm ischemia-reperfusion model after mice were fed ORY for seven days [142]. Studies have also shown that dietary restrictions exert protective effects in IRI in multiple organs, including the liver [143,144]. Miyauchi et al. found that twelve hours of fasting enhanced β-Hydroxybutyric acid expression, promoting the up-regulated expression of acetylated histone-3 and the activation of forkhead box protein O1 (FOXO1) and heme oxygenase 1 (HO-1) and enabling NF-κB and NLRP3 inactivation, thereby playing a protective role against LIRI-caused hepatocyte apoptosis and necrosis [145]. El-Sisi et al. found that octreotide could also play a protective role in LIRI by disrupting TLR4-mediated NLRP3 inflammasome activation and pyroptosis, and then inhibiting pyroptosis-triggered apoptosis [146].

An increasing number of studies have focused on the interaction between autophagy and the NLRP3 inflammasome in LIRI. Cao et al. found that pretreatment with 25 hydroxycholesterol (25HC)-activated mitophagy inhibited NLRP3 inflammasome activity in a rat LIRI model; the protective effects on the liver and the inhibitory effect of the NLRP3 inflammasome were attenuated after the inhibition of mitophagy, suggesting that the effect of 25HC pretreatment on LIRI may depend on upregulating mitophagy and inhibiting NLRP3 inflammasome activation [147]. Xue et al. demonstrated that lycopene promotes nuclear factor erythroid 2-related factor 2 (Nrf2)/ HO-1 pathway activation and further inhibits the NLRP3 inflammasome by enhancing pooled autophagy, thereby alleviating LIRI [148]. Wang et al. found that Eva1a inhibited NLRP3 activation, thereby alleviating LIRI by inducing Kupffer cell autophagy [149]. Zhang et al. found that transient receptor potential melastatin 2 knockdown attenuated hepatocyte oxygen-glucose deprivation/reoxygenation (OGD/R)-induced cell injury by enhancing autophagy and negatively regulated the NLRP3 inflammasome pathway. In addition, the use of INF39, an inhibitor of NLRP3, could increase cell viability and reduce cell apoptosis caused by OGD/R [150].

### 3.3. NLRP3 Inflammasome and Programmed Cell Death in CCl_4_-Induced Acute Liver Injury

Carbon tetrachloride (CCl_4_), an industrial solvent that causes the necrosis and apoptosis of hepatocytes in the centrilobular region, is a typical hepatotoxicant [151,152]. CCl_4_ is metabolized in the liver into trichloromethyl radical and peroxyl radical by cytochrome P450 enzyme (CYP2E1). The trichloromethyl radical can cause a peroxidation chain reaction to produce more toxic free radicals (such as superoxide and hydroxide anions), leading to free radical-mediated lipid peroxidation (membrane rupture, loss of membrane integrity) and thus liver cell damage [153,154]. Acute liver injury induced by CCl_4_ in animals is extremely similar to acute chemical liver injury in humans and is therefore widely used to investigate potential hepatoprotective strategies [155].

Salidroside (Sal), the active component of the plant salidroside, has many effects, including the inhibition of inflammatory responses. Zhang et al. found that Sal exerts protective effects against CCl_4_-induced ALI by reducing hepatocyte apoptosis, inhibiting oxidative stress, and reducing inflammatory responses by downregulating CYP2E1 expression and inhibiting NLRP3 inflammasome activation [156]. Zhao et al. found that Ginsenoside Rg1 ameliorated acute liver injury via autophagy and may be associated with the NF-κB/NLRP3 inflammasome signaling pathway [157].

### 3.4. NLRP3 Inflammasome and Programmed Cell Death in Alcohol-Induced Acute Liver Injury

Alcoholic liver injury is the most common cause of liver injury worldwide [158,159]. Drinking a lot in a short period of time can cause acute liver damage, even liver failure, and the mortality rate is as high as 44% [160]. Excessive drinking will activate the CYP2E1 in the liver, resulting in the excessive production and accumulation of ROS, and enhancing lipid peroxidation, leading to the swelling and disintegration of liver cells [161,162,163]. To date, only a few corticosteroid drugs are clinically available to treat alcohol-induced acute liver injury, and their efficacy remains unsatisfactory [164].

Liu et al. found that quercetin can reduce ROS production by inducing the expression of HO-1 and then downregulating the activation of NLRP3 inflammatory bodies and the secretion of proinflammatory cytokines to protect the liver from acute alcohol attack [164]. However, in the above studies, there was no further examination of whether NLRP3 inflammatory bodies could also regulate the PCD process.

### 3.5. NLRP3 Inflammasome and Programmed Cell Death in Con A-Induced Autoimmune Hepatitis

Autoimmune hepatitis (AIH), a complex hepatic inflammatory disease mediated by autoimmune reactions, can present in both acute and chronic forms [165]. Its clinical features are characterized by elevated serum transaminases, high γ- Hypogammaglobulinemia, autoantibody positivity, and interface hepatitis, with a predominance of lymphocytic and plasma cell infiltrates [166]. It occurs worldwide, with a relatively high incidence in European and American countries, and severe cases can rapidly progress to acute liver failure. Concanavalin A (Con A), a lectin extracted from concanavalin, causes liver injury via the tail vein injection of Con A and is a model of acute hepatitis because the damage it causes usually lasts only 48 h [6]. Most of its pathogenic features are similar to those of patients with clinical acute autoimmune hepatitis (AIH); as such, it has become one of the most commonly used research models for AIH [167,168].

Studies have shown that, in a Con A-induced ALI model in mice, the signal pathway involved in pyroptosis was significantly enhanced, which proved that pyroptosis was a key cell death event in AIH [169]. Phenethyl isothiocyanate (PEITC) is a widely distributed natural compound derived from the secondary metabolites of Brassicaceae plants; it has antimicrobial, anti-inflammatory, and antioxidant properties [170,171]. Wang et al., by constructing a Con A-induced acute immune liver injury model in mice, found that PEITC could reduce NLRP3 production and casp1 and GSDMD cleavage in the liver of ALI-suffering mice and directly interact with the cysteine at position 191 of GSDMD to inhibit hepatocyte pyroptosis, thereby exerting a significant hepatoprotective effect [172]. The work of Shi et al. showed that dimethyl fumarate promoted NLRP3 phosphorylation on Ser/Thr residues at specific sites of protein kinase A (PKA) and enhanced PKA signaling to inhibit NLRP3 inflammasome activation, pyroptosis, and IL-1 β Secretion, thereby treating Con A-induced AIH [173]. Deutsch et al. found that RIPK3 deletion was protective in Con A-induced AIH, whereas RIPK1 inhibition using Nec-1 exacerbated liver injury, and this effect was associated with increased hepatocyte apoptosis [128].

### 3.6. NLRP3 Inflammasome and Programmed Cell Death in LPS/D-GalN-Induced Acute Liver Injury

Lipopolysaccharide (LPS)/D-galactosamine (D-GalN) is a common hepatotoxic substance [174]. LPS/D-GalN combined with intraperitoneal injection can cause the diffuse necrosis of liver cells; this process is similar to the changes in liver pathology that occur after clinical acute viral hepatitis. Therefore, the animal model of ALI induced by LPS/D-GalN is widely used to explore the mechanism of clinical fulminant liver failure and test potential therapeutic drugs [175,176,177]. ALI induced by LPS/D-GalN overproduces TNF-α, IL-1β, IL-6, and other inflammatory cytokines, which together lead to the necrosis of liver cells and liver failure [178,179].

NLRP3-Inflammed Bodies and Autophagy Seem to Play an Important Role in LPS/D-GalN-Induced ALI. Several research groups have shown that natural products, such as Biochanin A, daphnetin, Licochalcone A, Mangiferin, and Salvia miltiorrhiza can reduce LPS/D-GalN-induced ALI by inducing autophagy to inhibit the activation of NLRP3 inflammasome [180,181,182,183,184]. Furthermore, Yang et al. found that maresin 1 could inhibit mitogen-activated protein kinase/NF-κB signaling and NLRP3 inflammasome-induced pyroptosis to ameliorate inflammation during LPS/D-GalN-induced ALI [185]. IL-1 exerts its effects through the cell surface interleukin-1 receptor type 1 (IL-1R1). Gehrke et al. found that LPS/D-GalN-induced acute liver injury was significantly attenuated in IL-1R1^Hep-/-^ mice, the expression of NLRP3 inflammasome and cell apoptosis were significantly inhibited, and the use of anakinra, an IL-1R antagonist, had the same effect [186]. S100 calcium-binding protein A9 (S100A9) is an important novel DAMP molecule that plays an important role in necroptosis. Bai et al. found that Paquinimod (an inhibitor of S100A9) significantly inhibited necroptosis and NLRP3 inflammasome activation in D-GalN/LPS-induced acute liver injury, thereby attenuating ALI [187].

## 4. Conclusions

The incidence of liver disease is increasing year by year and is the leading cause of death worldwide, and the immune response is central to almost all acute and chronic liver diseases. The NLRP3 inflammasome plays a critical role in host immune responses; However, its aberrant activation promotes the development of multiple chronic liver diseases (including viral hepatitis, alcoholic and nonalcoholic steatohepatitis, and liver cancer, among others) and ALI. ALI seriously threatens human health and quality of life and understanding how cell damage in the liver participates in the pathological process of ALI, no matter the cause, is the key to finding hepatoprotective measures. The role of the NLRP3 inflammasome in ALI has been widely recognized, and research on NLRP3 inflammasome activation has been intensive. A large number of studies have shown that NLRP3 inflammasome activation is inextricably linked to various types of PCD.

In this review, we summarize the pathogenesis of different types of ALI and discuss the role of the NLRP3 inflammasome and PCD therein (Table 1). The pathogeneses of various types of ALI are different, but a ROS signaling imbalance and an oxidative stress response seem to play a key role in all types. Several natural compounds ameliorate NLRP3 inflammasome activation and PCD in ALI by regulating oxidative stress and inhibiting the production of ROS. However, the specific targets of these compounds are not clear, and previous studies have been mostly based on animal models and cellular levels. The research is still lacking in vivo experimental validation, which limits the clinical use of these compounds in the treatment of ALI. The role of various targeted small molecule inhibitors in ALI has also been validated, suggesting that it may become a potential new drug in the clinic. In recent years, ferroptosis as one of the PCD forms is a research hotspot and was shown to play an important role in several diseases. However, this article mainly summarizes the interaction of NLRP3 inflammasome with apoptosis, necroptosis, pyroptosis, as well as autophagy, in ALI (Figure 3) and does not involve iron death in ALI, whether NLRP3 inflammasome can interact with ferroptosis and thereby participate in the pathogenesis of ALI needs to be further clarified and studied.

In conclusion, NLRP3 inflammasome and PCD are involved in the pathophysiology of ALI, and the rapid development of single-cell gene sequencing technology may help to further elucidate how PCD is coordinated with the activation of NLRP3 inflammasome, and how NLRP3 inflammasome regulates PCD. A full understanding of the link between them may not only improve our understanding of their pathogenesis but also facilitate the investigation of new targeted inhibitors, thereby providing a new strategy for the clinical treatment of ALI. 

## Figures and Tables

**Figure 1 ijms-24-03067-f001:**
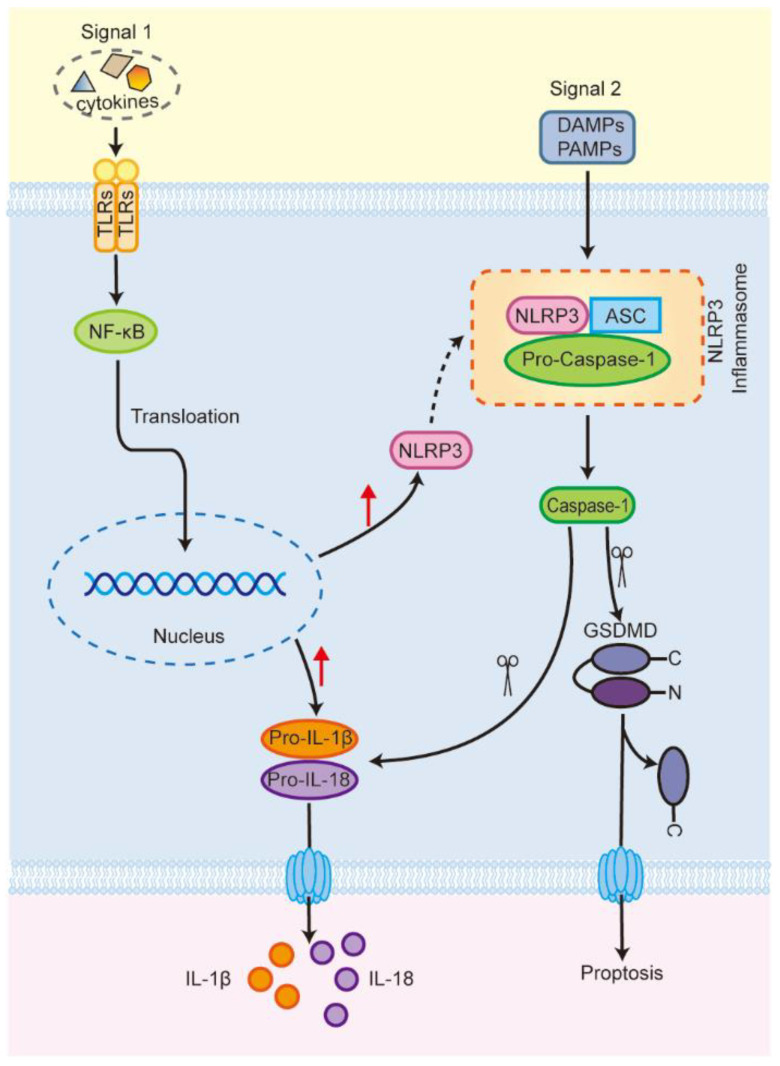
The activation process of NLRP3 inflammasome (Scissors represent clipping; red arrows represent rising). TLRs, toll-like receptors; NF-κB, nuclear factor kappa-B; NLRP3, nod-like receptor protein 3; Pro-IL-1β, pro-interleukin-1β; Pro-IL-18, pro-interleukin-18; ASC, apoptosis-associated spotted protein; Pro-caspase-1, pro-cysteine aspartate protease 1; GSDMD, gasdermin D.

**Figure 2 ijms-24-03067-f002:**
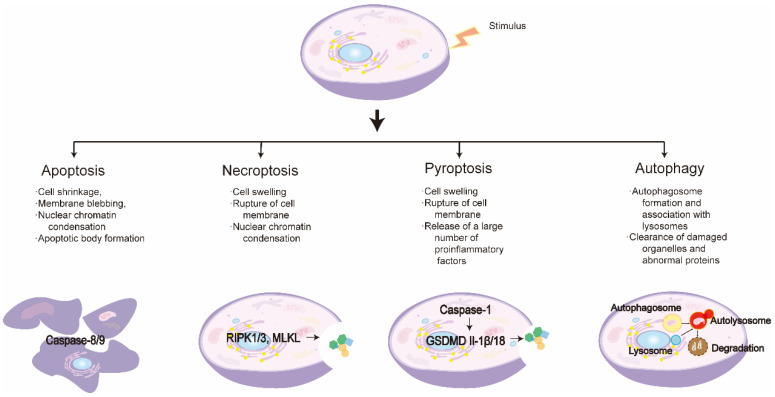
Characteristics of different programmed cell death pathways.

**Figure 3 ijms-24-03067-f003:**
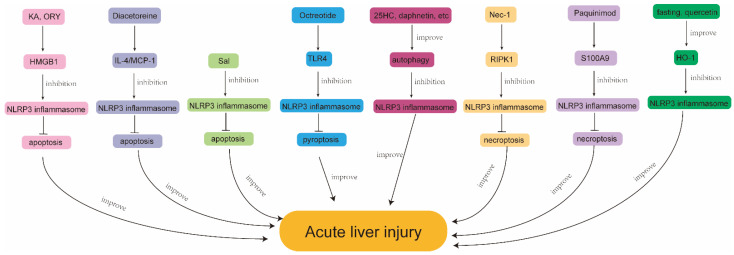
Mechanisms exerting protective effects by regulating NLRP3 inflammasome and PCD in ALI. KA, kaempferol; ORY, γ-Oryzanol; HMGB1, high mobility group protein B1; NLRP3, nod-like receptor protein 3; MCP-1, monocyte chemoattractant protein-1; Sal, salidroside; TLR4, toll-like receptor 4; 25HC, 25 hydroxycholesterol; Nec-1, Necrostatin-1; RIPK1, receptor interacting serine-threonine kinase 1; S100A9, S100 calcium binding protein A9; HO-1, heme oxygenase 1.

**Table 1 ijms-24-03067-t001:** Summary of the roles of NLRP3 inflammasome and programmed cell death in ALI animal models.

The Type of Acute Liver Injury	Animal	Experimental Model	The Role of NLRP3 Inflammasome and PCD	Reference
Acetaminophen-induced acute liver injury	C57BL/6 mice	Intraperitoneal injection of APAP (300 mg/kg)	Kaempferol inhibits HMGB1/TLR4/NF-κB signaling pathway and activation of the NLRP3 inflammasome, thereby alleviating apoptosis	[125]
	BALB/c mice	Intraperitoneal injection of APAP (400 mg/kg)	Diacetoreine downregulates NLRP3/caspase-1/IL-1β, IL-4/MCP-1, and TNF-α/ NF-κB proinflammatory signal pathway mediated oxidative stress, mitochondrial dysfunction, and apoptosis.	[126]
	C57BL/6 mice	Oral APAP (300 mg/kg)	Peroxiredoxin 3 inhibits NLRP3 inflammasome activation and prevents APAP-induced pyroptosis,	[127]
	C57BL/6 mice	Intraperitoneal injection of APAP (300 mg/kg)	Neurostatin-1, an inhibitor of RIPK1, alleviates APAP-induced liver injury by inhibiting the interaction between necroptosis and the NLRP3 inflammasome	[129]
Liver ischemia-reperfusion injury	C57BL/6 mice	70% liver ischemia-reperfusion model (1 h ischemia, 6 h reperfusion)	γ-Oryzanol protects the liver from I/R-induced inflammasome activation and apoptosis by inhibiting HMGB1/NLRP3.	[142]
	SD rats	70% liver ischemia-reperfusion model (30 min ischemia, 24 h reperfusion)	Octreotide plays a protective role in LIRI by disrupting TLR4-mediated NLRP3 inflammasome activation and pyroptosis	[146]
	SD rats	70% liver ischemia-reperfusion model (1 h ischemia, 3/24 h reperfusion)	25-hydroxycholesterol exerts protective effects by upregulating mitophagy and inhibiting NLRP3 inflammasome activation	[147]
CCl_4_-induced acute liver injury	C57BL/6 mice	Intraperitoneal injection of 10% CCl_4_ (500 μL/kg, diluted with olive oil)	Salidroside exerts protective effects against CCl_4_-induced ALI by reducing hepatocyte apoptosis and inhibiting the activation of NLRP3 inflammasome	[156]
	C57BL/6 mice	Intraperitoneal injection of 50% CCl_4_ (2 mL/kg, diluted with olive oil)	Ginsenoside Rg1 ameliorated acute liver injury via autophagy and may be associated with NF- κB/NLRP3 inflammasome signaling pathway	[157]
Alcohol-induced acute liver injury	Wistar rats	50% (*v*/*v*) ethanol by gavage (5 g/kg, three times)	Quercetin can induce the expression of HO-1 and then downregulate the activation of NLRP3 inflammasome	[164]
Con A-induced autoimmune hepatitis	ICR mice	Con A (10 mg/kg) was injected via tail vein	Phenethyl isothiocyanate can reduce NLRP3 production in the liver and directly interact with the cysteine at position 191 of GSDMD to inhibit hepatocyte pyroptosis	[172]
LPS/D-GalN-induced acute liver injury	C57BL/6 mice with Nrf2^−/−^ and wild-type	Intraperitoneal injection of LPS/GalN (30 μg/kg and 600 mg/kg)	Licochalcone A attenuates liver injury by inhibiting the activation of NLRP3 inflammasome via inducing autophagy	[180]
	C57BL/6 mice	Intraperitoneal injection of LPS/GalN (30 μg/kg and 600 mg/kg)	Maresin 1 inhibits mitogen-activated protein kinase/NF-κB signaling pathway and NLRP3 inflammasome-induced apoptosis	[185]

APAP, acetaminophen; HMGB1, high mobility group protein B1; TLR4, toll-like receptor 4; NF-κB, nuclear factor kappa-B; NLRP3, nod-like receptor protein 3; Caspase-1, cysteine aspartate protease 1; IL-1β, interleukin-1β; MCP-1, monocyte chemoattractant protein-1; TNF-α, tumor necrosis factor-α; RIPK1, receptor interacting serine-threonine kinase 1; ALI, acute liver injury; HO-1, heme oxygenase 1; GSDMD, gasdermin D.

## Data Availability

Not applicable.

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
