# Peer review of "The Role of the NLRP3 Inflammasome and Programmed Cell Death in Acute Liver Injury"

_ijms, 2023, doi:10.3390/ijms24043067_

Round 1
Reviewer 1 Report
My comments are:
-The author could better underline the link with immunity and tissue damage.
-Discuss in agreement and insert the following references in the text:
doi 10.1038/s41590-022-01386-w
doi 10.3390/ijms21062144
doi 10.1016/j.phymed.2022.154599
Reviewer 2 Report
Review for the manuscript
The role of programmed cell death and the NLRP3 inflammasome in acute liver injury
Dear Editor and authors,
Thank you for the opportunity to review this interesting manuscript. Although it is very interesting, I have some suggestions before it can be accepted for publication in IJMS.
Affiliation number 2 is “2.Institute of Stem Cell and Regenerative Medicine, School of Basic Medicine, Qingdao University, Qingdao, 6 China..”. Please, change for “2.Institute of Stem Cell and Regenerative Medicine, School of Basic Medicine, Qingdao University, Qingdao, 6 China.”
TITLE
I suggest including in the title that this is a review.
ABSTRACT
Since the title of this study is “The role of programmed cell death and the NLRP3 inflammasome in acute liver injury”, I suggest modifying the objective in lines 20-22 “In this review, we summarize the role of NLRP3 inflammasome activation and PCD in various types of ALI and analyze the underlying mechanisms to provide references for future relevant studies” for “In this review, we summarize the role of programmed cell death and the NLRP3 inflammasome activation in various types of ALI and analyze the underlying mechanisms to provide references for future relevant studies.” Other possibility is changing the title of the manuscript for “The role of NLRP3 inflammasome and programmed cell death in the acute liver injury, a review”. In my opinion, changing the title is better than changing the objective.
KEYWORDS
Please, remove bold from the first key-word.
INTRODUCTION
In line 29, we can see “…immune regulation, and detoxification.[1]”. The correct way is “immune regulation, and detoxification [1].”; In line 34 we see …”reperfusion liver injury, autoimmune liver injury, or viral liver injury;[2] viral causes…” The correct way is …”reperfusion liver injury, autoimmune liver injury, or viral liver injury[2]; viral causes…”. Please, correct along with all the text.
Please, include references published in 2022 in this section. There are so many in PUBMED.
In lines 42-43, we see, “The nod-like receptor protein 3 (NLRP3) inflammasome is the most widely studied of the inflammasomes It is composed”. Please, change to “The nod-like receptor protein 3 (NLRP3) inflammasome is the most widely studied of the inflammasomes. It is composed”.
Please, define abbreviations at the first time they appear in the text. For example, define NF-ĸB in line 47.
In Figure 1 we can find “cytonines”. Would it be “cytokines”?
Please, define all the abbreviations found in Figure 1 in its legend.
In lines 69-70 we see “physical, chemical, or mechanical stimuli (e.g., high temperature, high pressure, pH changes, shear forces).” Please, change for “physical, chemical, or mechanical stimuli (e.g., high temperature, high pressure, pH changes, and shear forces).”
In lines 78-80, please, revise the objective of the study according to my suggestion in the abstract section (see above).
In line 113-114 we see “In 1998, Vercammen found that fibrosarcoma cells induced a controlled form of necrosis in the absence of caspase.[43]”. Please, modify for “In 1998, Vercammen et al found that fibrosarcoma cells induced a 114 controlled form of necrosis in the absence of caspase.[43]”
In line 130, please, use italics for Shigella flexneri.
In line 196 we see “Yue Wang et al found that targeting…” Would it be “Wang et al found that targeting…”? See again in line 226. Please check this throughout the text. It is usual to refer to the author by his last name.
In table 1, the authors give the title "Table 1. Summary of the roles of NLRP3 inflammasome and programmed cell death in acute liver injury." However, it presents different compounds that act directly or indirectly on NLRP3 and, as a result, have some hepatic consequences. Therefore, if this table is to be kept, I suggest modifying its name so that it has a title that is consistent with its contents.
Please, include a legend for the abbreviations found in Table 1.
Please, put the Conclusions in a separate section.
Please, include the limitations of the study.
What are the future perspectives on the role of programmed cell death and the NLRP3 inflammasome in acute liver injury?
REFERENCES
Reference number 20 is missing.
Please, include more references published in 2022.
Reviewer 3 Report
This is very good and interesting review paper. But, still few things should be considered for its acceptance.
1. Topic regarding acute lung injury seems to be narrow. So, authors need to include more disease cases in liver including virus-induced hepatitis, fat liver-induced steatosis, liver cancer, and other chronic liver diseases.
2. Authors need to summarize and analyze gene expression pattern of PCD and NLRP3 inflammasome in human liver disease by using reported RNA seq data sets.
3. A lot of typo errors are found in the manscript (eg., .[25] should be [25]., CCL4, 4 should be subscript, L307, IL-1 β β, L324, /NF- κB ...etc).
4. Also, summary figure with signaling cascade and disease outcomes should be included in Conclusion section.
Reviewer 4 Report
The contribution of NLRP3 inflammasome activation in various types of in programmed cell death (PCD) including apoptosis and pyroptosis is reviewed using several preclinical models of ALI attempting reviewing mechanisms.
This is a review of different forms of ALI in preclinical animal models, which is of interest.
However, the review lacks an in depth analysis of the models and mechanisms in the different models and correlation with clinical ALI diseases. Further, the description of inhibitors is superficial, and focus essential on plant derived medicines, largely ignoring the established inhibitors, and relevant papers.
Therefore, I cannot support the current version for publication, specifically the following issues need to be considered:
1. Abstract: Need to mention which models or ALI addressed; more specific with preview of data, but reduce to 10 lines with a message.
2. Figures: need a detailed legend, Fig 1 typo: cytonines>cytokines
3. Inhibitors of NLRP3 and cell pathways: emphasis on established pharmacological inhibitors addressing the mode of action, and plant medicines as a subgroup.
4. Please add a discussion of the correlation of the models with the different clinical ALI forms, as well the predictability of the model such as the ConA and other models.
5. Figure 2: add a new figure with more details on autophagy.
6. Line 199: necerostatin>necrostatin; Line 319: NLRP3-inflammed bodies?
7. Add a list of abbreviations.
Round 2
Reviewer 2 Report
Dear authors,
Thank you very much for performing the corrections in your article.
With regards
Reviewer 3 Report
Authors have fully addressed and therefore it is now acceptable.
Reviewer 4 Report
I recommend the fully revised manuscript for publication.